# Less Can Be More: The Hormesis Theory of Stress Adaptation in the Global Biosphere and Its Implications

**DOI:** 10.3390/biomedicines9030293

**Published:** 2021-03-13

**Authors:** Volker Schirrmacher

**Affiliations:** Immune-Oncological Center Cologne (IOZK), D-50674 Cologne, Germany; V.Schirrmacher@web.de

**Keywords:** oxidative stress, low-dose radiation, metabolic switch, homeostasis, epigenetic memory, warburg effect, memory T cells, bone marrow, Nrf2, oncolysis, immunogenic cell death

## Abstract

A dose-response relationship to stressors, according to the hormesis theory, is characterized by low-dose stimulation and high-dose inhibition. It is non-linear with a low-dose optimum. Stress responses by cells lead to adapted vitality and fitness. Physical stress can be exerted through heat, radiation, or physical exercise. Chemical stressors include reactive species from oxygen (ROS), nitrogen (RNS), and carbon (RCS), carcinogens, elements, such as lithium (Li) and silicon (Si), and metals, such as silver (Ag), cadmium (Cd), and lead (Pb). Anthropogenic chemicals are agrochemicals (phytotoxins, herbicides), industrial chemicals, and pharmaceuticals. Biochemical stress can be exerted through toxins, medical drugs (e.g., cytostatics, psychopharmaceuticals, non-steroidal inhibitors of inflammation), and through fasting (dietary restriction). Key-lock interactions between enzymes and substrates, antigens and antibodies, antigen-presenting cells, and cognate T cells are the basics of biology, biochemistry, and immunology. Their rules do not obey linear dose-response relationships. The review provides examples of biologic stressors: oncolytic viruses (e.g., immuno-virotherapy of cancer) and hormones (e.g., melatonin, stress hormones). Molecular mechanisms of cellular stress adaptation involve the protein quality control system (PQS) and homeostasis of proteasome, endoplasmic reticulum, and mitochondria. Important components are transcription factors (e.g., Nrf2), micro-RNAs, heat shock proteins, ionic calcium, and enzymes (e.g., glutathion redox enzymes, DNA methyltransferases, and DNA repair enzymes). Cellular growth control, intercellular communication, and resistance to stress from microbial infections involve growth factors, cytokines, chemokines, interferons, and their respective receptors. The effects of hormesis during evolution are multifarious: cell protection and survival, evolutionary flexibility, and epigenetic memory. According to the hormesis theory, this is true for the entire biosphere, e.g., archaia, bacteria, fungi, plants, and the animal kingdoms.

## 1. Introduction

Hormesis describes a dose-response relationship to stressors with a low-dose stimulation and high-dose inhibition. The effect of the carcinogen dioxin on the development of breast cancer in rats serves as an example. In a low dose region, the frequency of tumors is greatly reduced when compared to no dioxin or to a high dose [1]. When testing the dose-response curve of chemotherapeutics, antibiotics, non-steroidal inhibitors of inflammation (NSAIDs), or toxins, a U-curve is seen with a reduction of toxic side effects at the nadir [2].

Hormesis is an evolutionary ancient biphasic dose-response of cells and it is a highly generalizable phenomenon [2]. A hormesis database from 2005 contains 5600 dose-response relationships over about 900 broadly diversified chemicals and physical agents [3]. Even hydrocarbons induce hormesis in biota at doses up to 100 times smaller than the toxicological threshold [4].

The linear non-threshold model (LNTM) extrapolates the late effects of high-dose exposure to ionizing radiation to the low-dose range and it is actually the cornerstone of current radiation protection policies. Advances in molecular and evolutionary biology, cancer immunology, epidemiological, and animal studies have cast serious doubts regarding the validity and reliability of LNTM [5]. Hormesis has emerged as a central concept of risk assessment for carcinogens and non-carcinogens. It has significant implications for clinical medicine [6].

This review, similar to a previous review on mitochondria [7], starts with evolution of this phenomenon on earth. Hormesis effects are described and explained in biochemical and molecular terms with special attention to the immune system. Clinical implications are exemplified in the fields of psychiatry, neurodegenerative diseases, cardiovascular diseases, metabolic syndrome, autoimmune diseases, and cancer. Hormesis effects are also described in plant cells with implications for agriculture. This review sheds light on the archaic origin of the adaptive stress response and elucidates its global validity.

## 2. Evolutionary Origin

### 2.1. The Beginnings

During billions of years, life on earth had to adapt to the changing environmental conditions. The anaerobic atmosphere gradually became enriched with oxygen (O_2_) due to the invention of photosynthesis by cyanobacteria. Thus, an antioxidant network evolved in bacteria to cope with the toxic effects of this new element in the atmosphere. The glutathion (GSH) system can exemplify this. After several hours of oxygen exposition of bacteria, a hormetic response can be seen at the transcriptional level by up-regulating nuclear factor erythroid 2-related factor (Nrf2)-mediated expression of the enzymes involved in GSH synthesis. In phototrophic bacteria, adaptations also eventually occurred at the epigenetic and genetic level [8].

*A chemo defense system*. A chemo defense system has been proposed to have evolved very early protecting organisms against toxic substances. Mechanisms that are involved, for example, lipophilic compounds, hydrophilic compounds, oxidants, acidosics, genotoxics, and metals. By analogy with the later evolving immune defense system, the chemo defense system can be characterized, as follows: partial immaturity of the young, inducibility, non-specificity, and specificity [9].

Hormesis via quorum sensing (QS) receptors. The hydrogen ion concentration (H^+^ or H_3_O^+^) of an aqueous solution (logarithmic measure via pH)) was demonstrated to affect the hormesis response of bacteria. The pH profiles of certain compounds affected the luminescence response of the Vibrio qinghaiensis sp.-Q67 [10]. The compounds displaying hormesis bound more easily to the α subunit of luciferase than to the ß subunit [10]. Luminescence in *Allivibrio fischeri* bacteria was studied to investigate hormetic mechanism of sulfonamides (SAs) on bacterial QS cell-cell communication. It was suggested that SAs acted on quorum sensing LitR proteins to change their active forms. This then induced hormetic effects on luxR (QS signal receptor, [11]), and thereby affected the luminescence [12]. SAs triggered time-dependent hormetic effects on growth of Escherichia (*E. coli*) bacteria over a time span of 24 h. It was reported that SAs bind with adenylate cyclase at a low dose and with dihydropteroate synthase at a high dose. New insights revealed a role of energy source in this hormesis system [13].

*Protection against UV light.* The protective effects of the monoterpenes camphor, eucalyptol, and thujone were studied in *E. coli* K12 bacteria. The results were consistent with a hormesis response. At a low dose, the agents protected the bacteria against UV-induced mutagenesis and carcinogen 4-nitroquinoline-1-oxide (4NQO)-induced DNA strand breaks. Similar effects were seen with DNA repair proficient mammalian Vero cells [14].

*Ionizing radiation hormesis.* Radiation hormesis and toxicity were investigated with luminous marine bacteria. Bioluminescence intensity was used as physiologic parameter to study the effects of exposure to alpha- and beta-emitting radionuclides (americium-241, uranium-235 + 238, and tritium). Three successive stages of response were detected: (1) the absence of effects (stress recognition); (2) activation (adaptive response); and, (3) Inhibition (suppression of physiological function, i.e., radiation toxicity) [15].

*Glycohormesis.* Experiments with cells from yeast strains revealed that hormesis enables the cells to handle accumulating toxic metabolites during increased energy flux [16]. Reactive carbonyl (RCS) and reactive oxygen (ROS) species caused cellular damage through the production of advanced glycation endproducts (AGEs) and oxidative stress. Preconditioning with methylglyoxal (MG) led to changes in metabolism and activated the protein quality control system (PQS). It was concluded that, next to mitohormesis, there also exists glycohormesis. A direct link between metabolic and proteotoxic stress was suggested. Specific therapeutic interventions, e.g., the manipulation of chaperone systems, might open new fields for drug development and the treatment of diseases involving increased RCS and ROS levels, such as diabetes mellitus (DM) and neurodegenerative diseases [16].

*Fasting stress and differentiation induction.* Dietary restriction stress can induce the reproduction cycle in slime molds. Slime molds (Dictyostelium) belong to a branch that separated from archaea before the fungus kingdom Mycota. A proteome-based eukaryotic phylogenetic tree from 2005 is based on six archaebacterial proteomes: Malaria parasite (*Plasmodium faciparum*), green alga (*Clamydomonas reinhardtii*), rice (*Oryza sativa*), maize (*Zea mays*), fish (*Fugus rubripes*), and mosquito (*Anopheles gambiae*). It revealed that slime molds belong to a branch that is designated as Amoebozoan [16]. Dictyostelim discoideum is an important source of structural and functional information. In the case of dietary restriction stress, single cells aggregate and induce stalk-cell differentiation via polyketide differentiation-inducing factor-1. On top of the stalk, in the fruit body, further differentiation steps occur [17]. Thus, fasting induced signals for the reproduction cycle, including a change from a unicellular to a multicellular organism.

*Fasting-induced autophagy.* The aim of another investigation was to test an anti-oxidative cellular protection effect of fasting-induced autophagy as a mechanism for hormesis. Marine snails (*Common periwinkle*, Littoria littoria) were used as an animal model. These snails were deprived of algal food for seven days to induce an augmented autophagic response in their hepatopancreatic digestive cells (analogues of hepatocytes). Fasting significantly increased cellular health in terms of lysosomal membrane stability, reduced lipid peroxidation, and lysosomal/cellular triglyceride. It reduced potentially harmful lipofuscin, an age-pigment of proteinaceous aggregates [18].

*Fasting, endoplasmatic reticulum (ER) stress, and proteostasis*. Studies in worms, such as Caenorhabditis (*C. elegans*), demonstrated that dietary restriction improved proteostasis and increased the life span. The mechanism worked through ER hormesis. The unfolded protein response (UPR) of the ER helped to maintain proteostasis in the cell [19].

*Responses to environmental stress.* In *C. elegans*, environmental stresses were shown to induce inheritable survival advantages via germline-to-soma communication. Animals that were subjected to various stressors during developmental stages exhibited increased resistance to oxidative stress and proteotoxicity. The increased resistance was transmitted to the subsequent generations that were grown under unstressed conditions through epigenetic alterations. In the parental somatic cells the insulin/insulin-like growth factor (IGF) signaling effector DAF-16/FOXO and the heat-shock factor HSF-1 mediated the formation of epigenetic memory. This was maintained through the histone H3 lysine 4 trimethylase complex in the germline across generations. The elicitation of memory required the transcription factor (TF) SKN-1 (homology of mammalian Nrf2) in somatic tissues [20].

The positive effects of mild stress on ageing and lifespan have been mainly studied and documented in cells from worms (*C. elegans*) and insects (*Drosophila melanogaster*) [21]. Mild stress, including hypergravity [22] and mild cold stress [23], protects and improves animal performance. Hormesis is known by multiple names: preconditioning, conditioning, pretreatment, cross tolerance, and adaptive homeostasis [24]. Dietary restriction (DR), fasting (FA), and cold exposure (CE) are hormetic stress models [25]. Rapid stress hardening (RCH) is the fastest acclimatory response to low temperature known and it is a key adaptation for coping with thermal variability, especially for ectotherms such as crustaceans, terrestrial arthropods, amphibians, and reptiles. It was originally reported in 1987 in a Science paper. When the flesh fly Sarcophaga crassipalpis was exposed to cold shock at −10 °C for 2 h, this caused >80% mortality. However, when only 30 min. of exposure to 0 °C preceded the same cold shock, mortality decreased to <50% [26]. Molecular mechanisms that were associated with RCH across species revealed biological processes, such as allelic variation, transcription (e.g., heat shock proteins, cryoprotectant synthesis), translation (e.g., calcium signaling, redox balance), post-translational modifications (e.g., p38/MAP kinase, mRNA surveillance), and biochemical changes (e.g., cryoprotectant accumulation and membrane fluidity) [26].

Over time, hormesis has become recognized as a fundamental concept in biology. It affects, for example, toxicology, microbiology, medicine, public health, and agriculture [27].

Table 1 provides an overview of the main features, mechanisms, and effects of part II.

### 2.2. Nrf2 and Its Role in Anti-Oxidative and Anti-Inflammatory Cellular Responses

A protein that is homologous to the transcription factor Nrf2 already existed in the worm *C. elegans*. The Nrf2 signaling pathway in mammals plays a pivotal role in controlling the expression of antioxidant genes and exerts anti-inflammatory functions. Molecular details have recently been elucidated [28]. Under normal homeostatic conditions, in the cytosol of mammalian cells, Kelch-like ECH-associated protein 1 (Keap1) homodimerizes with an E3 ligase. This complex (Keap1-Cul3-RBX1) interacts with the Keap1 binding domain of Nrf2 and it leads to Nrf2 ubiquitination and degradation [28]. 

Certain cystein residues of Keap1 are highly reactive and susceptible to covalent modifications by ROS, RNS, H_2_S, and other electrophiles and by ER stress. S-sulfenylation, S-nitrosylation and S-sulfhydration of these critical cysteins causes conformational changes of Keap1. This, together with phosphorylation of Nrf2 by protein kinases, promotes the dissociation of Nrf2 and its stabilization. This is followed by Nrf2 nuclear translocation, heterodimerization with small Maf proteins (sMaf), and binding to the anti-oxidant response elements (AREs), leading to the transcription of ARE-driven genes, such as heme oxygenase-1 (HO-1) [28]. 

In addition to this Nrf2 signaling pathway, Nrf2 interferes with the nuclear factor kappa-light-chain-enhancer of activated B (NFκB) pathway that initiates inflammation. Inflammation is a response to a variety of biological threats, such as infection by pathogens and tissue injury. The first step is the detection of an infection signal and/or damaged tissue signal. Such signals are mediated by pathogen-associated molecular patterns (PAMPs) and damage-associated molecular patterns (DAMPs). These exogenous and endogenous molecular patterns are recognized via pattern recognition receptors (PRRs), which are expressed by immune cells. Toll-like receptors (TLRs) or inflammosomes activate specific immune signaling pathways result in the activation of NFκB. 

The response to TLR activation starts with the phosphorylation of the NFκB/IkB complex and the dissociation of NFκB from IκB. This is followed by the translocation of NFκB to the nucleus and the induction and transcription of genes coding for pro-inflammatory cytokines (e.g., interleukin-1-beta (Il-1ß), IL-6, tumor necrosis factor-alpha (TNF-α), and others. These cytokines then recruit immune cells, such as monocytes and neutrophils, at the site of infection or tissue damage. Their activation leads to the generation of reactive oxygen and nitrogen species (ROS, RNS), which cause damage of macromolecules, such as proteins and DNA. Under normal physiological conditions, such as wound healing, restoration blocks any further neutrophil recruitment and then re-establishes tissue homeostasis.

However, in chronic “inflammation”, the risk of cellular damage is multifold. Sustained inflammatory response causes tissue injury. The release of chemokines and prostaglandins recruits further inflammatory cells, resulting in a respiratory burst and elevated oxidative stress. The activation of transcription factors, such as NFκB and Nrf2, are key components of inflammation signaling cascades and oxidative stress responses. The above described Nrf2/HO-1 axis, activated by ROS, can interfere with NFκB in inflammation. This includes the inhibition of NFκB activation, blocking the degradation of IκB-α, degradation of NFκB, and inhibition of NFκB nuclear translocation. The latest insights into these complex regulatory interactions have recently been excellently reviewed [28]. Table 2 provides an overview of Nfr2 and its role in anti-oxidative and anti-inflammatory cellular responses.

## 3. Low-Dose Radiation (LDR) Mediated Hormesis Effect in the Immune System

High-dose radiation (HDR) usually results in immune suppression. In contrast, low-dose radiation (LDR) modulates a variety of immune responses and reveals properties of immune hormesis. Hormetic effects include cells of innate immunity and cells of adaptive immunity [29].

### 3.1. LDR and Innate Immunity

LDR can enhance the activity of NK cells by stimulating cell proliferation and promoting their cytotoxic function [30]. This was associated with the p38/MAPK (mitogen-activated protein kinase) signaling pathway [30]. It was documented that LDR programs macrophage differentiation towards M1 polarization. The cells were positive for inducible nitric oxide synthase (iNOS) and they could orchestrate effective T cell immunotherapy [31]. LDR was also described to affect oxidative burst in stimulated macrophages [32]. Reports on the effects of LDR on dendritic cells (DCs) are conflicting. The irradiation of DC precursors with 0.5 Gy in vitro did not influence lipopolysaccharide (LPS) induced surface marker expression or cytokine profile [33]. In contrast, it was reported from another study that 0.05 Gy pre-treated DCs exhibited the highest proliferation-inducing capacity on T cells, and then augmented the production of interleukin (IL)-2, IL-12, and IFN-γ [34]. 

### 3.2. LDR and Adaptive Immunity

LDR was reported to enhance the response of CD4+ helper T cells, in vitro and in vivo [35]. Similarly, an enhanced CD8+ cytotoxic T cell (CTL) response was reported following LDR treatment [36]. The molecular mechanisms likely involve activated survival/signaling proteins (e.g., NFκB, p38/MAPK, and c-Jun N-terminal kinase (JNK)) [37]. LDR also led to an increase in production of immune enhancing cytokines (IL-2 and IL-4) and a decrease in the production of a major immunosuppressive cytokine (transforming growth factor-ß1 (TGF-ß1)) [37]. Furthermore, CD markers (e.g., CD3, CD2, CD4, and CD28) became upregulated by LDR [38]. With regard to regulatory T cells (Tregs), some studies showed that they were markedly decreased in mice or rats following LDR [39]. IL-10, the most relevant cytokine mediating Treg suppressor activity, was also downregulated by LDR [40].

LDR can also affect many aspects of B cell behavior. It can modulate B cell differentiation through the activation of NFκB and the induction of the cell differentiation molecule CD23 [41]. LDR can increase global genomic DNA methylation [42] and promote a metabolic shift from oxidative phosphorylation (OXPHOS) to aerobic glycolysis. This leads to increased radiation resistance in human B cells [43].

In conclusion, low-dose therapeutic irradiation, as well as background irradiations (e.g., radon spas), is beneficial rather than destructive from an immunological point of view [43].

## 4. Other Hormetic Effects in the Immune System

The immune system is continuously influenced by hormetic effects of environmental compounds (e.g., chemicals), physical influences (background irradiations, major change of temperature), or medical (drugs) and food interactions [44]. Low-level ROS activates the main cellular antioxidant pathways (e.g., thioredoxin (TRX) and GSH) as well as their transcriptional regulator Nrf2 [45].

A hormesis effect that was elicited by a carcinogen (dioxin) was described in 2003 [46]. The experiment tested the effect of dioxin on the development of breast cancer in rats. Figure 1 illustrates the results. In a low-dose region, the frequency of tumors was greatly reduced when compared to no dioxin or to a 100-fold higher dose of the carcinogen. The shape of the curve is described as U or J or inverted bell. The mechanisms behind this effect have not been elucidated. The activation of the immune system in the low-dose region is one possibility.

### 4.1. Protection by Immunological Memory

Immunological memory is a cardinal feature of the adaptive immune system. Bodywide immune surveillance is based on circulating cells, including central, effector, and peripheral memory T cells (MTCs). Local immune surveillance is based on tissue resident MTCs. At steady state, MTC homeostasis is under the control of various cytokines, transcription factors, and metabolic fuels [47].

Mammals evolved in the face of fluctuating food availability. The effects of transient dietary restriction (DR) on the immune system have been studied in mice. Under DR, MTCs collapsed in secondary lymphoid organs and accumulated in the bone marrow (BM). Glucocorticoids coordinated the BM response It involved profound remodeling while adopting a state of energy conservation involving adipogenesis. Interactions of CXCR4 chemokine receptor with its ligand CXCL12 and BM trophic factors contributed to T cell accumulation. In this way, the BM protected and optimized immunological memory. MTC homing to BM under DR was associated with enhanced protection against infections and tumors [47].

### 4.2. Increase of Longevity and Tissue Protection by Macrophages as Hormesis Effects against Biological Threats

A study with fruit flies (Drosophila melanogaster) reported that pathogenic fungus spore challenge increases the longevity and fecundity, but results in reduced anti-fungal immune function [48]. Thus, the beneficial effects of low level exposure to toxins and other stressors may not necessarily, and under all conditions, help the immune system [48].

Another study demonstrated a hormesis mediated dose-sensitive shift in macrophage activation patterns. The activation or polarization of macrophages to pro- or anti-inflammatory states evolved as an adaptation to protect against biological threats. The study demonstrated: (1) many pharmacological, chemical, and physical agents can mediate a shift between pro- and anti-inflammatory activation states; and, (2) these shifts display biphasic dose-response relationships that are characteristic of hormesis. The study also revealed that preconditioning similarly mediates tissue protection by the polarization of macrophages. However, in this case, the direction was towards an anti-inflammatory phenotype [49].

The microbiome also influences hormesis. A review of the literature revealed influences on oncogenesis and therapy. Microorganisms were documented to have the ability to perturb risks of cancer and enhance hormesis after irradiation [50].

### 4.3. Hormetic Effects on the Immune System by Sportive Exercise

Sportive exercise can affect the innate/inflammatory responses. Such effects are primarily mediated by the sympathetic nervous system (SNS) and/or by the hypothalamic-pituitary-adrenal (HPA) axis. Stress hormones from the adrenal glands (catecholamines and glucocortioides) play an important role [51]. Table 3 provides an overview of the main features, mechanisms, and effects of parts III and IV.

## 5. Clinical Implications

### 5.1. Low Stimulatory Effects of Toxic Compounds

Formaldehyde (FA) is the first example. This is an indoor environmental pollutant, classified as a carcinogen. Human K562 leukemia cells and human 16HBE brochial epithelial cells were exposed to different concentrations of FA. At low concentrations, FA promoted proliferation of both cell types by inducing key molecules of cell division like CyclinD-cdk4 (cyclin-dependent kinase 4) and E2F1 (E2F transcription factor 1). In addition, key molecules of the Warburg effect were increased: pyruvate kinase isozyme M2 (PKM2), glucose, glucose transporter 1 (GLUT1), lactic acid, and lactate dehydrogenase A (LDHA) [52].

The second example is hydrogen peroxide. LDR (<100 mGy) mediates nanomolar release of hydrogen peroxide (H_2_O_2_) as a stable product of water radiolysis. H_2_O_2_ has recently been recognized as a central redox signaling molecule. LDR utilizes known molecular master switches, such as Nrf2/Keap1 or NFκB, to promote adaptive resistance. It has been proposed that LDR mediates its hormetic effects via H_2_O_2_ signaling [53].

### 5.2. Psychiatry

Lithium (Li) is one compound with a hormetic effect in psychiatry. Salts of Li are -carbonate, -acetate, -sulfate, -citrate, -orotate, and -gluconate. They are used as medicines in psychiatry to treat manic episodes and depressions, therapy resistant schizophrenia, and other indications. New studies in the fruit fly Drosophila suggest that Li promotes longevity. The life-extending mechanism that is involved the inhibition of glycogen synthase kinase-3 (GSK-3) and activation of the transcription factor Nrf-2. High levels of Nrf-2 activation conferred stress resistance, while low levels additionally promoted longevity [54].

### 5.3. Neurodegenerative Diseases

It is now accepted that neuroinflammation is a common feature of neurological diseases. Cytosolic inflammasomes are multiprotein complexes. They are part of the innate immune system. They activate proinflammatory caspases, thereby leading to the activation of proinflammatory cytokines (e.g., interleukin (IL)-1b, IL-18 and Il-33). These cytokines promote neuroinflammation and brain pathologies. The best characterized receptor family in Alzheimer’s disease (AD) is the nucleotide-binding oligomerization domain-like receptor family, pyrin domain-containing-3 (NLRP3) inflammasome. A recent review introduced the concept of hormesis and presented possible mechanisms and applications to neuroprotection. It proposed the potential utility of the nutritional antioxidants sulforaphane and hydroxytyrosol [55].

The neuropeptide receptor pituitary adenylate cyclase-activating polypeptide receptor 1 (PAC1-R) mediates neuroprotective activity. It was reported recently that H_2_O_2_ exerts a hormesis effect on the promoter activity of this receptor [3]. PAC1-R mediates well-known neuroprotective, neurotrophic, and neurogenesis effects and is an important drug target for neurodegenerative diseases. The study might help to further clarify the physiological effect of low-dose ROS on the nervous system [56].

Curcumin, a polyphenol compound from the rhizome of Curcuma longa Linn, is another neuroprotective antioxidant hormetin. It is commonly used as a spice to color and flavor food. The common denominator for its potential protective role in neurodegenerative disorders is the enhancement of the cell stress response, mainly due to the heme oxygenase-1 system [57]. Curcumin mediates an intricate crosstalk between mitochondrial turnover, autophagy, and apoptosis [58]. Curcumin was recently reported to regulate ROS hormesis by favoring mitochondrial fusion/elongation, biogenesis, and improved function in rodent muscle cells [57]. The curcumin safety profile imposes a careful analysis of the risk/benefit balance prior to proposing chronic supplementation. Similar conclusions can be drawn from the proposed hormesis via Ginseng to achieve neuroprotective effects [59].

Isoliquiritigenin (IsoLQ) is a flavonoid with antioxidant properties and is an inducer of ER stress. It was shown that IsoLQ pretreatment of a kidney cell line induced ER stress-mediated hormesis [60]. Other agents inducing a hormetic dose response are chloroquine [61] and green tea with its principal constituent (-)-epigallocatechin-3-gallate (EGCG) [62]. Both of the agents have been demonstrated to enhance a spectrum of neuroprotective responses.

Neurological injury is a frequent cause of morbidity and mortality from general anesthesia and related surgical procedures. Cold-preconditioning is a procedure for neuroprotection. One study used hippocampal slice cultures to investigate neural immune signaling via cytokines that are derived from microglia [63].

Transcranial brain stimulation with low-level light/laser therapy (LLLT) is another strategy to modulate neurobiological function in a nondestructive and non-thermal manner. The mechanism of action of LLLT is based on photon energy absorption by cytochrome oxidase, the terminal enzyme in the mitochondrial respiratory chain. LLLT can provide neuroprotection and cognitive enhancement by facilitating mitochondrial respiration, with hormetic dose-response effects and brain region activational specificity [64].

### 5.4. Cardiovascular Diseases (CVD)

Low levels of ROS were shown to decrease the susceptibility of neonatal rat ventricular myocytes to anoxia/reoxygenation injury and it also caused profound protection in an in vivo mouse model of ischemia/reperfusion. Higher levels of ROS resulted in a progressive alteration of intracellular Ca^2+^ homeostasis and mitochondrial function in vitro, leading to dysfunction and death. ROS levels were regulated by the mitochondria-targeted redox cycler MitoParaquat (MitoPQ). The data support a hormetic model, in which low levels of ROS are cardioprotective, while higher levels are cardiotoxic [65].

A recent study investigated the role of NLRP3 inflammasomes in cardiac aging by comparing NLRP3 -knockout and wild-type mice. The absence of NLRP3 prevented age-related mitochondrial dynamic alterations in cardiac muscle. In wild-type mice, melatonin supplementation had an anti-apoptotic action in addition to restoring Nrf2-antioxidant capacity and improving mitochondria ultrastructure altered by aging [66].

### 5.5. Metabolic Syndrom

Metabolic syndrom (MetS) includes obesity, insulin resistance, hypertension, and atherogenic dyslipidemia. It is associated with an increased risk of type 2 diabetes mellitus (T2DM), myocardial infarction, and stroke. MetS have been estimated to affect 20–30% of adults worldwide [7]. T2DM confers an excessive risk for CVD. It is preceded by dysfunction in vascular reactivity. (-)-Epicatechin (EPICAT), a plant compound and known vasodilator, modulates mitochondrial redox levels in vascular models of oxidative stress. The data showed that EPICAT acts in a dose-dependent manner, demonstrating hormesis [67].

Intermittent metabolic switching (IMS) has been proposed by Mark Mattson to maintain neuroplasticity and brain health [68]. Based on animal model studies, he suggested that switching between time periods of negative energy balance (short fasts and/or exercise) and positive energy balance (eating and resting) can optimize general health and brain health. ß-hydroxybutyrate (BHB) is a ketone, generated from fatty acids during fasting and extended exercise, which functions as a cellular energy source. As the signaling molecule BHB induces the expression of brain-derived neurotrophic factors. The mammalian target of rapamycin (mTOR) is a serine/threonine protein kinase that plays a pivotal role in stimulating cellular protein synthesis and suppressing autophagy when nutrients (glucose and amino acids) are plentiful [57]. Biochemical pathways that are involved in the metabolic switch connect the organs liver, gut, and brain, and the cells hepatocytes, adipocytes, neurons, and astrocytes. Fasting and exercise lead to a glucose-to-ketone switch (bioenergetic challenge) and cellular stress resistance (molecular recycling and repair pathways). Eating, resting, and sleeping lead to a ketone-to-glucose switch (recovery period) and to cell growth and plasticity pathways (mitochondrial biogenesis, synaptogenesis, and neurogenesis) [68].

A hormetic response of mitochondria (mitohormetic response) has been proposed as part of the cytoprotection mechanism of berberine (Ber). This is a major bioactive compound that is extracted from plants (Coptis chinensis), which has anti-diabetic effects. Ber mainly accumulates in mitochondria. It targets enzymes and other proteins that are associated with the electron transfer chain (ETC) or with mitochondrial DNA (mtDNA) to disrupt energy homeostasis and induce translation stress. This stress can induce mitohormetic responses via: (1) ROS-mediated redox pathway; (2) AMP/ATP-induced AMPK pathway; (3) NAD+/NADH-mediated Sirtuins pathway; and, (4) the UPRmt pathway to regulate and maintain mitochondria homeostasis for the ability of cells to adapt to adverse circumstances [69].

### 5.6. Autoimmune Diseases

Autoimmune diseases result from a hyperactive immune system attacking normal tissue. The regulatory effect of LDR on the immune system seems to depend on the immune microenvironment of the body. Repeated LDR was reported to significantly inhibit the osteoclastic activity in patients with rheumatoid arthritis [70]. The anti-inflammatory effect of LDR was suggested to be an important mechanism by which LDR affects autoimmune disease [71]. UV irradiation also affected the immune system and occurrence of autoimmune diseases [72]. LDR inhibited the expression of proinflammatory cytokines, upregulated the proportion of Tregs, and reduced the production of autoantibodies [73]. In autoimmune situations, LDR inhibited the transformation of immature DCs (imDCs) into mature DCs (mDCs) and induced the differentiation/polarization of macrophages into M2. Altogether, LDR seems to be capable of regulating the negative effects of immune dysregulation in autoimmune diseases [29].

### 5.7. Acute Respiratory Distress Syndrome (ARDS)

Because LDR can induce an anti-inflammatory phenotype, it has been suggested as a possible treatment option for COVID-19-induced acute respiratory distress syndrome [74]. The two-phase immune responses that are induced by this new virus have been explained [75]. The lessons learned from SARS and MERS epidemics have been summarized [76]. Lessons from other pathogenic viruses have also been described [77]. 

The Front Line COVID-19 Critical Care Alliance (FLCCC), a clinical expert panel, developed a treatment protocol based on the core therapies methylprednisolone, ascorbic acid, thiamine, heparin, and co-interventions (MATH+) for hospitalized patients [78]. Recently, ivermectin, an anti-parasitic medicine with potent anti-viral and anti-inflammatory properties against COVID-19 was added to the list. An update from Dec. 18, 2020, reviews the emerging evidence demonstrating the efficacy of ivermectin in the prophylaxis and treatment of COVID-19 [79].

### 5.8. Multidrug Interaction

In elderly patients with chronic diseases, each organ often affected is treated by one or several different drugs. This is because clinical education is organ oriented and prioritized by pharmacotherapy. Polypharmacotherapy is clinically manifested by a reduction in the effectiveness of pharmacotherapy. Perhaps, interventions that are based on dysregulated mitochondria could help to reduce multidrug usage [7].

A high consumption of drugs, combined with their presence in the environment (e.g., antibiotica in meat), raises concerns regarding consequences. A recent study analyzed individual and drug mixture acute toxicity. It tested the pharmaceuticals diazepam, metformin, omeprazole, and simvastatin. The test organism was the bacterium *Allivibrio fischeri*. In terms of individual toxicity, omeprazole was the most toxic agent. When the toxicity of mixtures was tested, synergisms, antagonisms, and hormesis effects were observed, most probably due to byproduct formation. This work points to the urgent need for more studies that involve mixtures, since chemicals are subject to interactions and modifications, can mix, and potentiate or nullify, the toxic effect of each other [80]. Table 4 provides an overview of the main features, mechanisms, and effects of part V.

## 6. Is Less More in Cancer Therapy?

### 6.1. Historic Aspects

The development of cancer therapies has gone a long way: from surgery to adjuvant radiotherapy, chemotherapy, and hormone therapy. Many dogmas dominated in certain time periods and were then proven wrong. Examples include: (1) the dogma of radical, ultra-radical, and supra-radical surgery that dominated from 1891 to 1981 [81]; and, (2) the dogma of aggressive high-dose chemotherapy (CT) that dominated the years 1980 to 2000 [82,83].

An example where less aggressive therapies are possible: early-stage breast cancer patients are treated by surgery plus adjuvant CT. In 2016, Cardoso et al. published a paper in the New England Journal of Medicine regarding a new 70-gene signature test (MammaPrint) to determine a genomic risk as an aid to treatment decisions in early-stage breast cancer [83]. A randomized Phase III study enrolled 6693 women with early-stage breast cancer and determined their genomic risk and their clinical risk. Women with high clinical risk and low genomic risk of recurrence based on Mammaprint received no CT. Their five-year survival rate without distant metastases was similar to that of patients treated with CT. It was concluded that approximately 46% of women with early-stage breast cancer who are at high clinical risk might not require CT [84]. This is an example where less aggressive therapies are possible, at least in patients with a low genomic risk.

There is growing evidence that a radical re-evaluation of the mode of action of chemotherapeutic agents and ionizing radiation is required in light of advances in immunology and hormesis research. The concept of hormesis that was applied to cancer therapy suggests that different modes of action of therapeutic modalities exist at different doses. Thus, a change of paradigm would be necessary. In the case of CT, this may mean changing from the maximum tolerated dose concept to low intermittent (e.g., metronomic) therapy. In radiation therapy, lower doses and accurate stereotactic targeting might enable antigen-releasing (immunogenic) doses of radiation to be delivered to the tumor with a sparing of surrounding normal tissue. Coupled with emerging immunotherapies, the future of cancer treatment may consist in more localized debulking surgery, repositioned CT, and radiotherapy in combination with immunotherapy and targeted therapies [85].

### 6.2. Hormetic Aspects of Targeted Therapies, Oncolytic Viruses and Cancer Vaccines

#### 6.2.1. Hormetic Aspects of Small Molecule Inhibitors (SMIs)

The mammalian target of rapamycin (mTOR) regulates, among others, aerobic glycolysis in carcinomas. It regulates the metabolism of glucose, amino acids, fatty acids, lipids, and nucleotides. Small molecule inhibitors (SMIs) might be suited to target cancer-associated molecules that are associated with mTOR and glycolysis [86]. SMIs must optimally fit into an enzyme’s active site to inhibit its functional (e.g., tyrosine kinase) activity. This means that the dose-response relationship is non-linear and it has an optimum at a molecular enzyme to inhibitor ratio of 1:1.

#### 6.2.2. Hormetic Aspects of Antigen Recognition by the Immune System

Antigen–antibody interactions, as studied by Paul Ehrlich’s toxin-anti-toxin precipitation studies, revealed, at the equivalence point, a molecular key to lock ratio of 1:1. Similar rules guide cognate T cell interactions with antigen. There are three participants in the molecular recognition of antigen by T cells: an antigenic fragment (1. peptide) that forms a complex with a presenter molecule (2. MHC protein), and this complex is recognized by a recognition molecule, the antigen-specific T cell receptor (3. TCR). Only an optimal fit between an antigenic peptide-MHC (pMHC) complex and the corresponding TCR, in the presence of costimulatory signals, initiates a T cell response. 

That the dose-response to vaccines is not linear has also to do with cell–cell interactions. The initiation of a T cell response requires T cell interaction with a professional antigen-presenting cell (APC), such as a DC. Only cognate interactions lead to a response. Cognate T-APC interactions occur when a T cell with a TCR for an antigen (e.g., A) finds an APC expressing A. Such interactions can occur in vaccination-site draining lymph nodes, in the spleen and in the BM [87,88]. Cognate interactions between memory T cells from the BM and tumor antigen (TA)-presenting DCs revealed bi-directional cell stimulation, survival, and antitumor activity in vivo [89].

#### 6.2.3. Hormetic Aspects of Oncolytic Viruses and Cancer Vaccines.

The roles of hormesis seem to also apply to the mechanisms of oncolytic virus (OV) therapy. Systematic research on oncolysis, the selective destruction of tumor cells by viruses, began in the 1960s. Using ascites tumor cells growing in the peritoneal cavity of mice, it was found that neither low nor high virus doses were effective. Only an intermediate dose caused macroscopically visible oncolysate. The dose-response curve was bell-shaped, like in hormesis [90]. 

In the following, the focus will be on oncolytic Newcastle disease virus (NDV). The author studied this avian paramyxovirus since the 1980s. The goal was not oncolysis, but anti-tumor vaccination. The first impressive findings were obtained in the aggressive ESb mouse lymphoma model in 1985. (1). Post-operative vaccination with irradiated ESb cells caused no protective effect against micrometastatic disease, so that all mice died within 2–3 weeks. In contrast, post-operative vaccination with NDV-infected irradiated ESb cells (ESb-NDV vaccine) caused approximately 50% long-term survival. Like with oncolysis, an intermediate dose of virus had to be used to obtain an immunogenic vaccine [91]. (2). Further research revealed that ESb-NDV vaccine caused in vitro an augmented tumor-specific CD8+ mediated CTL response in ESb-immune spleen cells in comparison to stimulation with uninfected ESb vaccine [92]. Similar augmented effects were obtained when analyzing a CD4+ mediated T helper cell response [93]. When titrating the virus to tumor cell ratio, it was found that an optimal response required approximately 10 virus particles per tumor cell [94]. (3). In the years between 1990 and 2008 (the time-point of the authors retirement at the German Cancer Research Center in Heidelberg, Germany), translational research allowed for creating a human virus-modified vaccine homologous to ESb-NDV. It was designated as Autologous Tumor cell Vaccine that was modified by NDV infection (ATV-NDV) [95]. Delayed-type hypersensitivity (DTH) skin reactivity was used as first immunogenicity assay in Phase I studies in cancer patients. These studies revealed that an optimal virus to tumor cell ratio was seen at about 10 virus particles per tumor cell [96]. Findings (1)–(3). can be well interpreted as a hormesis effect. Similar dose responses existed for mice and human.

The results that were obtained with the vaccine ATV-NDV were true for the lentogenic NDV strain *Ulster*, which has only monocyclic replication capacity in tumor cells. In the DC-based vaccine IO-VAC^R^ [97,98] that is being used since 2015 at the Immune-Oncological Center Cologne (IOZK) Germany, the patient-derived NDV oncolysate is obtained with a mesogenic NDV strain that shows multicyclic replication capacity. In this case, the virus to tumor cell ratio can be titrated down to 1 or 0.1 virus particles per tumor cell. More than one million tumor cells are normally used in a tumor cell-based (ATV-NDV) or DC-based (IO-VAC^R^) vaccine.

The concept of individualized, multimodal immunotherapy (IMI), which was developed at IOZK, combines immunogenic cell death (ICD) treatment via NDV with modulated electrohyperthermia (mEHT) and IO-VAC^R^ DC vaccination. A recent data analysis of 70 treated adult patients that were from Glioblastoma multiforme (GBM) revealed that IMI, in combination with maintenance chemotherapy, provides a strategy towards improving the overall survival rate [99]. In the same review, the concept of randomized controlled immunotherapy clinical trials for GBM has been questioned and challenged [99].

The Rho GTPase Rac1 plays an important role in GBM cell migration and invasion. Interestingly, Rac1 is also a target of NDV infection. It is involved in viral entry, during syncytium induction, and upon actin reorganization [100,101]. NDV-induced syncytium formation triggers autophagy, which is mediated through the activation of the AMPK (energy-sensing AMP-activated protein kinase)-mTORC1-ULK1 (autophagy-initiating protein kinase) pathway [102]. This network plays a role in autophagy and in maintaining cellular energy and nutrient homeostasis.

Solid tumor microenvironments contain regions of hypoxia, in which a distinct transcription factor (i.e., hypoxia inducible factor (HIF)) is active. A velogenic NDV strain was applied in order to compare the oncolytic effect against a clear cell carcinoma line under normoxic and hypoxic conditions. It was found that NDV could break resistance to hypoxia. Hypoxia even augmented oncolytic activity, regardless of the HIF levels in the cells [103]. 

Resistance to therapy is a major obstacle to cancer treatment. NDV was reported to have the potential to break resistance not only to hypoxia, but also to chemo- and radiotherapy, to apoptosis, to tumor-necrosis-factor-related apoptosis-inducing ligand (TRAIL), and to immune checkpoint blockade [104].

NDV pre-treatment of cancer patients before vaccination has an immune conditioning effect. Immune cells are primed towards a type I interferon response via signaling through cytoplasmic RIG-I receptor and through plasma membrane expressed type I interferon receptor [105]. Upon NDV infection in vitro, human DCs become programmed within 18 hrs. into DC1 polarization. A sophisticated study revealed that the antiviral response of human DCs to NDV infection is highly reproducible and dictated by a choreographed cascade of 24 transcription factors leading to the upregulation of 779 genes [106]. 

The dose-response to vaccines is not linear, as mentioned above. A determination of maximally tolerable dose, as required from toxicology, is meaningless in immunology. With regard to the vaccine ATV-NDV, the first study of post-operative vaccination of breast cancer patients defined a dose above one million cells and below five million cells and cell viability of the irradiated cells above 50% as high quality parameters, based on patient survival [94]. Competence and polarization of the patient’s immune system are other parameters of importance, and possible negative effects of combinatorial treatments. Future optimizations should investigate vaccination schedules, delivery routes, and biomarkers. The side effects of systemic cancer therapies is another important point. A recent comparative analysis revealed that cancer vaccines and oncolytic viruses exert profoundly lower side effects in cancer patients than other systemic therapies [107].

Further evidence for a hormetic response by the NDV infection of tumor cells was obtained in 2002 with a paper entitled: “Dendritic cells pulsed with viral oncolysate potently stimulate autologous T cells from cancer patients” [108]. The paper reports that primary operated breast cancer patients contain in their BM cancer-reactive memory T cells (MTCs). These were stimulated in vitro with DCs that were pulsed with lysate from the breast carcinoma cell line MCF-7 or with lysates from NDV-infected MCF-7 cells. An ELISPOT test revealed that the latter in comparison to the former induced a significantly increased interferon gamma (IFN-γ) response. The supernatants from such cultures contained increased titers of interferon alpha (IFN-α) and interleukin 15 (IL-15). Further danger signals apart from foreign viral RNA in the cytoplasm of the tumor cells, MALDI mass spectrometry, Western blots, FACS cytometry, and ELISA tests were employed to analyze potential. NDV infection of MCF-7 cells resulted in a number of differences in protein expression by Western blots. MALDI mass spectrometry of prominent proteins revealed a heat-shock protein (HSP), which, upon NDV infection, became phosphorylated: HSP27 [108].

As molecular chaperones, HSPs constitute a large family of conserved proteins that play a key role in intracellular protein homeostasis. They are involved in protection against various stress factors. Members of different HSP families also become localized on the plasma membrane of cancer cells and they could become interesting new targets for cancer therapy [109].

A new study from 2020 revealed that expression of human HSP27 in yeast cells extends replicative lifespan and uncovers a hormetic response. HSP27 is a small heat shock protein that modulates the ability of cells to respond to heat shock and oxidative stress. It functions as a chaperone independent of ATP, and it participates in the proteasomal degradation of proteins. In cancer cells, it confers resistance to CT, in neurons, HSP27 has a positive effect on neuronal viability in models of Alzheimers’s and Parkinsons’s disease [110].

### 6.3. Low-Dose T Cell Triggering and Cytotoxic Effector Function

Upon APC contact, the mobile T cells scan the APC’s cell surface for the presence of exactly fitting pMHC complexes. This scanning for maximal key-lock fits might explain the low-dose detection limit for T cell triggering. Four pMHCs per TCR cluster are sufficient for triggering. The vast majority of the about 10,000 peptides of an APC in vivo are normal self peptides that do not elicit a response [88].

Nature has invented mechanisms of tolerance to limit anti-self peptide auto-immune reactivity. During the maturation of T cells in the thymus, negative and positive selection steps lead to central tolerance. This ensures that only those mature T cells leave this organ, whose TCRs recognize self MHC molecules in association with non-self peptides [111].

The interactions of a CTL specific for a given antigen A with a tumor target cell expressing A depend on the CTL to target cell ratio and they can be quantified in vitro in respective cytotoxicity assays. Such titration curves are not linear. Cognate interactions between CD4+ T helper cells and CD8+ CTL precursor cells are important for the generation of a long-lasting protective immune response. Therefore, CD8+ peptide vaccines, including T helper cell epitopes, are superior to those without helper epitopes [112].

#### 6.3.1. mRNA-Based Vaccines

mRNA-based vaccines against COVID-19 have been developed in 2020 within less than a year by several start-ups in cooperation with large pharma companies. The principles have been worked out already 20 years ago [113,114]. Two studies used RNA coding for ß-galactosidase (ß-gal) as model TA. One study demonstrated that polycationic peptide protamine-protected RNA and naked RNA can be used in vivo in mice to elicit specific CTLs and antibodies [113]. The other study [114] introduced an additional two innovative procedures for further optimization of RNA vaccination. (1) Use of the mouse ear pinna as vaccination site and (2) the use of self-replicating infectious RNA. The mouse ear pinna had been shown before to be a vaccination site superior to other commonly used sites [115]. The self-replicating RNA vaccine made use of the Semliki Forest virus replicase to drive RNA expression of the lacZ gene coding for ß-gal. A 100-fold lower dose of polynucleotide was sufficient in comparison to a lacZ DNA vector for achieving a protective response against lacZ-transfected tumor cells with self-replicating RNA [114].

Thus, mRNA-based vaccines are not new. One of the two studies from 20 years ago was performed in Tübingen (Germany) in the laboratories of Hans-Georg Rammensee, the other study was performed in Heidelberg (Germany) at the German Cancer Research Center in the laboratories of the author of this review.

#### 6.3.2. Peptide-Based Vaccines

It has been reported that peptide vaccination can lead to enhanced tumor growth through specific T-cell tolerance induction [116]. Additionally, combinatorial cancer therapies might negatively impact T cell responses, as shown in a phase III peptide vaccination study combined with the tyrosine kinase inhibitor sunitinib [117]. Such results demonstrate that considerable knowledge and care are needed to find optimal conditions for eliciting T cell responses against cancer by peptide-based vaccines.

A recent review discusses relevant issues to be solved before the implementation of peptide vaccinations in the standard treatment of tumor patients, e.g., target antigen selection, adjuvant choice, vaccination schedule, and delivery routes. In addition, the clinical treatment concepts must be clarified. Three different strategies are being discussed: (1) stratification; (2) warehouse-based personalization; and, (3) individualization [118].

### 6.4. Mitohormesis, Macrophages and Case Reports

Adverse factors, such as genetic mutations, hypoxia, nutritional deficiencies, and drug toxicity, result in the accumulation of unfolded proteins in the ER, causing ER stress. To survive, cancer cells trigger the unfolded protein response UPR. Non-coding RNAs (ncRNAs) play important roles in regulating protein translation and adaptation to adverse environments [119]. Mitochondria also are capable of exerting a UPR response (UPR mt). A recent paper discussed the role of the UPRmt in maintaining cancer cells in the mitohormetic zone to provide selective adaptation to stress [120].

*Methylglyoxal and advanced glycation end products*. Metabolic reprogramming towards aerobic glycolysis in cancer cells favors the production of MG and AGEs. It was reported that MG exerts a hormetic effect that is defined by a low dose stimulation and a high dose inhibition of tumour growth. The use of MG scavengers could switch tumors from growth to death [121].

Many chemotherapeutic treatments induce cell death by increasing intracellular ROS concentration. Persistent drug stimulation leads tumor cells to stimulate a hormetic process, by which the cells exhibit a biphasic response to the drugs used. In this framework, ß3-adrenoreceptors (ß3-Ars) fit with an antioxidant role in cancer. Selective ß3-AR antagonists, such as SR59230A, lead cancer cells to increase ROS concentrations, thus inducing cell death [122].

A hormetic relationship to outcome has been reported with regard to tumor-associated macrophages in classical Hodgkin lymphoma (HL). Seventy-six samples of HL were subjected to immunohistochemical double staining using CD68 or CD163 macrophage specific monoclonal antibodies. Because the MYC oncogene is supposed to control expression of M2-specific genes in macrophages, the immunohistochemistry also involved a reagent to detect MYC. Cases with highest numbers of macrophages usually showed the worst Disease-free Survival (DFS) and Overall Survival (OS). In most of the samples, intermediate numbers of macrophages were associated with a better outcome than very low or very high numbers [123].

A high expression of Nrf2 was found to be associated with increased tumor-infiltrating lymphocytes and cancer immunity in ER-positive/HER2-negative breast cancer. This was based on in silico analyses in 5443 breast cancer patients from several large patient cohorts. High Nrf2 tumors were highly infiltrated by immune cells (CD8+, CD4+, and DCs) and stromal cells (adipocytes, fibroblasts, and keratinocytes [124]. In contrast, the negative effects of Nrf2 expression have been reported for glioblastoma [125] and lung adenocarcinoma [126]. Tumor entities presenting oncogenic activation of Nrf2 were found to be associated with drug resistance and immune evasion [125,126].

The chapter will be finished with two case reports from Shuji Kojima and colleagues from the Department of Radiation Biosciences, Tokyo University of Science, Chiba, Japan. The first deals with treatment of cancer and inflammation (ulcerative colitis) by low-dose ionizing radiation. The three case reports support the clinical efficacy of low dose radiation hormesis in patients with these diseases [127]. The second publication reports four cases of radon therapy as a primary or an adjuvant treatment for different types of cancer [128]. It is recommended to perform clinical trials to determine the best radon concentration for the treatment of different types of cancers and in different stages of progression [128]. Table 5 provides an overview of the main features, mechanisms, and effects of part VI.

## 7. Hormesis Effects in Plants

Hormesis is a well-known phenomenon not only in the animal kingdoms, but also in plants. The mechanisms involved are still poorly understood. A recent study investigated the role of oxidative stress, auxins (plant hormones), and photosynthesis in corn that was treated with cadmium (Cd) or lead (Pb). In corn seedlings, the gas exchange and the chlorophyll α fluorescence, as well as the content of chlorophyll, flavonol, auxin, and H_2_O_2_, were measured. The common features of the hormetic stimulation of shoot growth by heavy metals were an increase in the auxin and flavonol content and the maintenance of H_2_O_2_ at the same level as the control plants [129].

Nanoparticle silver (AgNP) treatment of maize has a beneficial, possibly hormetic, effect on the plants roots. However, a recent analysis of the maize rhizospere revealed significant multiple unintended effects of nanosilver use on corn. Specifically, the microbial rhizome community structure and expressed genes of both prokaryotic and eukaryotic microorganisms was studied. Diversity analysis indicated a significant decrease in richness. Among the phylum bacteria, some groups (e.g., Chloroflexi and Planctomycetes) decreased significantly, while other groups (e.g., Acidobacteria, Bacteroidetes, and Proteobacteria (Alpha and Gamma)) were increased in response to nanosilver exposure. Among the phylum fungi, an increase in abundance was observed, including potentially phytopathogenic groups. Certain species from the genus Diplodia are causal agents of stalk and ear rot in maize, and this genus showed a 5.5 fold increase under nanosilver exposure. It was concluded that the disruption of natural biocontrol could cause phytopathogen increase. Compromised nitrogen cycling, possible phytopathogen selection, and plant hormesis effects were detected via metatranscriptome analysis. In the long term, this could turn out to be negative to crop productivity and ecosystem health [130].

Additionally, some herbicides, like glyphosate, 2,4-D and paraquat, at low dose, exert a hormetic response. When ROS are produced, H_2_O_2_ acts as a signaling molecule that promotes cell walls malleability, allowing for inward water transport causing cell expansion [131,132].

Silicon (Si) is a beneficial element that has been proven to influence plant responses, including growth, development, and metabolism in a hormetic manner. After oxygen, Si is the second most abundant element in the Earth’s crust. It covers up to 32% of the litosphere. It is found as silicates and Si minerals, combined with oxygen or elements, like aluminum (Al), manganese (Mg), calcium (Ca), sodium (Na), iron (Fe), and potassium (K). In plants, Si can only be absorbed as monosialic acid (Si(OH)_4_). It is then transported and mainly deposited in the cell apoplast. Si concentrations in plants fluctuate between 0.1% and 10% of the total dry mass. Seven of the 10 most produced crops in the world are Si accumulators, and these respond positively to Si applications. These crops include rice, wheat, barley, sugarcane, soybean, and sugar beet [133].

A recent study [133], performed with pepper plants (Capsicum annuum L.) revealed hormetic dose-response effects of Si on growth and concentrations of chlorophyll, amino acids, and sugars during the early developmental stage. Si was supplied as calcium silicate (CaSiO_3_) in the nutrient solution. It was applied at four levels: 0, 60, 125, and 250 mg L^−1^. Si differentially affected plant growth and metabolism, depending on the concentration applied. Si might act as a signal to promote amino acid remobilization to support the increased demand of nitrogen during grain development. Si interacts with key components of plant signaling systems. This includes binding to the hydroxyl groups of proteins involved in cell signaling. It can also act as a signaling modulator by interacting with cationic co-factors of enzymes influencing stress responses.

As sessile organisms, plants have evolved unique mechanisms that enable them to face the complexity of environmental changes. Future recommendations to agronomists will include Si applications to fields that are deficient in the element. The rapid pace of global climate change leads to new challenges for agriculture and food production [133]. Table 6 provides an overview of the main features, mechanisms, and effects of part VII.

## 8. Archaic Environmental Stress Response as an Example of Hormesis

The environmental stress response (ESR) was originally identified in yeast. It is characterized by a rapid and transient transcriptional response composed of large, oppositely regulated gene clusters. In a global program, such as the ESR, a large fraction of the transcriptome is rapidly and transiently reprogrammed in response to stress. The ESR is induced in response to a variety of stressful conditions, which suggests a core transcriptional response [134]. 

Characteristics of a core stress response, such as cross-stress protection and adaptation, are observed among eucaryotes and bacteria. Cross-stress protection has been reported in response to temperature and osmotic stress in bacterial model organisms, such as *Bacillus subtilis*, *Listeria monocytogenes*, and *Escherichia coli*. Adaptive responses to stress have been reported across fungal species. Recent phylogenetic evidence suggests that eukaryotes may have originated from within the archaeal branch of the tree of life [135]. Therefore, it was of great interest to find out whether archaea can also exert an ESR.

The basal transcriptional machinery in archaea, like that of eukaryotes, consists of the general transcription factor (TF) B (homologous to TFIIB), TATA binding protein, and an RNA-polymerase (Pol) to initiate transcription. Stress-responsive TFs in archaea resemble those of bacteria. Such TFs can bind directly to a signaling ligand (e.g., metal, sugar, and metabolite) to activate or repress transcription. A recent study analyzed global transcriptional programs in archaea [135]. Halobacterium (Hbt. salinarum) served as an archaeal model organism. It was found that this archaeal species exhibits a eukaryote-like ESR. This fulfilled the typical four criteria [135].

1. Global, stereotypical transcriptional reprogramming. In yeast (S. cerevisiae) cells that served for comparison, 868 genes were involved in the ESR, comprising more than 14% of the genome. Of those genes, 283 were induced and 585 repressed (iESR and rESR, respectively). In cells of *Hbt. salinarum*, the iESR contained 724 genes and the rESR 276 genes. This suggests that, like in eukaryotes, large-scale transcriptional coordination of seemingly disparate cellular processes may also be active in archaea.

2. Induced and repressed genes that are enriched for distinct functions. In yeast, iESR encode a variety of protective and damage repair processes, including carbohydrate metabolism, protein folding and degradation, defense against oxidative stress, intracellular signaling and others. rESR genes of yeast are enriched for functions that are associated with optimal growth, including translation (e.g., ribosome synthesis and processing) and RNA Pol I-and III-dependent transcription. For iESR genes in *Hbt. salinarum* cells, only genes without a functional classification were significantly enriched, which suggested a need for further research. The cituation was clearer for rESR genes in these cells. Repressed genes were enriched for functions involved in large and small ribosome subunit biogenesis and assembly, peptide biosynthesis, metabolic processes, ATP metabolic process, and regulation of translation. The repression of genes that are involved in ribosome biosynthesis and translation was shared between the two cell types.

3. The duration and magnitude of the transcriptional response dependent on the intensity of stress. In yeast cells, extreme heat shock (25 °C to 37 °C) elicited a greater transcriptional response than lower heat shock. In *Hbt. salinarum* cells the duration and magnitude of response was tested with regard to oxidative stress. This was exerted by the redox cycling agent paraquat. Low concentrations were 0.25 mM, high concentrations 4 mM. HhHigh-dose treatment mounted a higher magnitude change when compared to low-dose. Gene expression returned to nearly pre-treatment levels after 150 min of exposure.

4. The induction of the transcriptional response specific to stress exposure. In yeast cells, a reciprocal environmental shift (37 °C to 25 °C) caused a rapid transition to basal expression levels without the peak seen under the condition of 3. Similarly, *Hbt. salinarum* cultures recovering from oxidative stress returned rapidly to basal expression levels without exhibiting ESR-like transcriptional characteristics. Similar, non-reciprocal dynamics were observed upon treatment with hydrogen peroxide.

In conclusion, upon sensing changes in the surrounding environment, *Hbt. salinarum* exhibits transient transcriptional dynamics that are characterized by the induction and repression of large portions of the genome (criteria 1 and 2). This response is specific to stressful conditions and sensitive to the magnitude of stress (criteria 3 and 4). It was further suggested that TrmB family proteins are candidate regulators of the ESR in archaea [134].

Stressors that were tested across archaeal species were specific to the respective niche of the extremophile of interest, including hypo-osmotic shock for halophiles, temperature extremes for hyperthermophiles, and others. A common trend was that genes encoding core cellular processes required for rapid growth are repressed during stress. In particular, the repression of translation has been reported across species and stress conditions. Thus, it is appropriate to talk about global genome-wide transcriptional programs as conserved features of the ESR. Table 7 provides an overview of the main features, mechanisms, and effects of part VIII.

## 9. Global Aspects

Oxygen, approximately two billion years ago, a waste product of photosynthetic cyanobacteria, induced oxidative stress. Gradually, the production of ROS became a driver of physiological and pathological processes. Low-level ROS play an important role as redox-signaling molecules in a wide spectrum of pathways that are involved in the maintenance of cellular homeostasis and regulating key transcription factors (e.g., Nrf2/Keap1, NFκB/IκB, AP-1, p53, and HIF-1) [132].

Melatonin, in coordination with the circadian rhythms, is involved in stress adaptive responses [136]. This hormone is produced in animals by the pineal gland and in plants under stress. Substantial evidence was provided of a melatonin-induced biphasic dose-response relationship. This showed similarities to those of broad toxicological and pharmacological hormesis literature. This example from chronobiology means, for instance, for medicine, that finding the right dose is not all, the right time point for drug application is also important.

Melatonin may act as a conditioning agent protecting organisms against subsequent health threats within a hormetic framework. The incorporation of melatonin-induced hormesis in research protocols has the potential to enhance the treatment of neuropsychiatric diseases and cancers. It may also help in the protection against environmental stress in plants and to increase plant productivity [136].

Hormesis has been suggested to promote evolutionary change and the rescue of phenotypic plasticity [137]. Genetic recombination, nonlethal mutations, activity of transposable elements, or gene expression are some of the molecular mechanisms through which hormesis might enable organisms to maintain, or even increase, evolutionary fitness in stressful environments. These mechanisms span the tree of life from plants to vertebrates. The inheritance of epigenetic memory provides the offspring with survival advantages [20].

Three complex biochemical systems operate for cellular homeostasis and they are involved in hormesis: the proteasome (P), the endoplasmic reticulum (ER), and mitochondria (M). These components have been united in the PERM hypothesis [138]. The PERM hypothesis can explain via hormesis the beneficial role of many xenobiotics, either trace metals or phytochemicals, which are spread in the human environment and dietary habits. These exert their actions on the mechanisms that underlie cell survival (apoptosis, autophagy, cell cycle regulation, DNA repair, and turnover) and stress response. They act on the energy balance, redox system, and macromolecular turnover. If PERM-mediated control is offline, impaired, or dysregulated, reactive species (RCS, ROS, RNS) and stressors could have a negative effect. That seems to be the case in metabolic syndrome, degenerative disorders, chronic inflammation, and cancer [7]. Ionized calcium might play a role in maintaining the correct rhythm of PERM modulation [138].

Another recent review emphasizes that environmental, physical, and nutritional hormetins lead to the stimulation and strengthening of the maintenance and repair systems in cells and tissues. Exercise, extreme temperature (heat or frost), and irradiation are examples of physical hormetins. The molecular mechanisms of the hormetic response include modulation of: (1) transcription factor Nrf2 activating the synthesis of glutathione and the subsequent protection of the cell; (2) DNA methylation and epigenetics; and, (3) microRNA [139].

Rewriting the history of toxicology and pharmacology is an appeal to change of paradigm. This is due to the fundamental biological basis of environmental hormesis. Low doses of environmental agents have recently been reported to induce autophagy, a critical adaptive response that essentially protects all cell types. Hormesis can also be transgenerational via epigenetics. The reviewer appeals to stakeholders in toxicology and pharmacology to re-examine the process of risk assessment, with the goal of optimizing public health, rather than simply avoiding harm [140]. Table 8 provides an overview of the main features, mechanisms, and effects of part IX.

## 10. Discussion

Hormesis is a theory of non-linear dose-response relationship. It can explain many, but not all, phenomena about how cells respond to low-dose exposure of stressors. Not all responses are beneficial to host survival. One example should elucidate this. Tumor dormancy is an important, but not yet well understood, phenomenon in cancer research. Using dormancy models of lung and ovarian cancer, it was recently described that modified lipids that are derived from stress-activated neutrophils lead to reactivation of dormant tumor cells. Stress hormones cause rapid release of proinflammatory S100A8/A9 proteins by neutrophils. These induce the activation of myeloperoxidase, resulting in the accumulation of oxidized lipids. Upon release from neutrophiles, these lipids upregulate the fibroblast growth factor pathway. This causes tumor cell exit from dormancy and the formation of new tumor lesions [141]. Thus, stress factors can also exert detrimental effects.

The developing immune system serves as a novel target for disruption by environmental chemicals and drugs. The effects can significantly influence later-life health risks. Optimal mitochondrial function is critical during embryonic development. Mitochondria play a key role in early signaling cascades and epigenetic programming [142]. The right nutrition is very important in the perinatal period because of epigenetic imprinting. Neonatal brain injury has been linked to an iron-dependent form of cell death (ferroptosis) that is characterized by enhanced lipid peroxidation [143]. 

Mitochondria are sensitive targets of environmental toxins, potentially even at levels that are considered to be safe under current regulatory limits. Twenty-four anthropogenic chemicals were recently tested for their effects on embryonic oxygen consumption rate (eOCR). Each chemical, depending upon the concentration, resulted in a unique eORC response profile. Non-monotonic dose response effects and mitochondrial hormesis were detected with some chemicals. The authors conclude that mitochondrial responses to chemicals are highly dynamic and warrant careful consideration when determining the mitochondrial toxicity of a given chemical [144].

The range of postnatal health risks linked to developmental immunotoxicity (DIT) is influenced by the natural progression of prenatal to neonatal development. Pregnancy imposes a Th2-bias in utero. This produces a delay in the acquisition of Th1 functional capacity in the newborn. Because hormesis has been shown to be an important factor in the modulation of the adult immune system, more research is required to understand potentially opposing the dose-response effects of xenobiotics for the immune system of the fetus, neonate, and juvenile. A direct linkage between immune dysfunction and chronic disease has become abundantly apparent in recent years [145].

This review has reported the positive hormetic effects of transient dietary restriction (DR) and intermittent metabolic switching (IMS). This is in contrast to the permanent stress of starvation and malnutrition worldwide. In 1989, 23 leading hunger experts in their Bellagio Declaration defined four achievable goals to overcome hunger: eliminate famine death; end hunger in half of the world’s poorest households; reduce by half malnutrition of mothers and small children; and, eradicate iodine and vitamin A deficiencies [146]. In 2009, the WHO and UNICEF recommended a transition to WHO growth standards to identify wasting for children aged from six to under 60 months. This has led to the evolution of a worldwide logistic system to provide emergency food aid. Malnutrition worldwide includes a spectrum of nutrient-related disorders that are major public health problems: intrauterine growth retardation, protein–energy malnutrition, iodine deficiency disorders, vitamin A deficiency, iron deficiency anemia, and overweight and obesity [147]. Infants aged under six months are often excluded from nutrition surveys. However, the fact is that, in developing countries, large numbers of infants under six months are wasted. A data analysis from 2011, using WHO standards, revealed that about three million infants under six months were severely wasted and 2.5 million moderately wasted worldwide [148].

The principles of hormesis have entered the field of physical exercise and athletic performance training. The effects of exercise on the innate immune system are influenced, among others, by stress proteins, such as HSP72. Regular exercise can induce immuno-neuroendocrine stabilization in persons with deregulated inflammatory and stress feedback by reducing the presence of stress hormones and inflammatory cytokines. Nevertheless, biomedical side effects of exercise need to be considered [149]. According to evolutionary biology, organisms may exhibit growth under stress, a phenomenon that is designated as antifragility. For coaches and their athletes, a key question is how to design training conditions to help athletes develop the kinds of physical, physiological, and behavioral adaptations underlying antifragility. A recent review discusses how to determine optimal stress loads for antifragility in climbing. It includes individualized load-response profiles [149].

Recent developments in low-dose effects research provide a novel means in environmental toxicology and ecotoxicology to improve the quality of hazard and risk assessment [150]. Herbicide hormesis is commonly observed at subtoxic doses of herbicides and other phytotoxins. However, it can cause undesired effects in which weeds are unintentionally exposed to hormetic doses in adjacent fields [151]. There may also be stimulatory effects of low concentrations of herbicides as environmental contaminants spread over estuaries and lakes. One example are the phytoplankton blooms. A recent hormetic research on *Microcystis aeruginosa* and *Selenastrum capricornutum* suggests that the blooms were triggered by herbicides and involved cytochrome b_559_, ROS, and NO [152]. It was recommended that, in environmental toxicology and ecotoxicology, rethinking is necessary to provide more reliable estimates of risk assessment and optimize health [150].

Homeostasis describes a system of balance of a cell with respect to energy and environment. Mitochondria play an important role in maintaining homeostasis [7]. Hormesis, which should not be mixed-up with homeopathy, describes the biochemical mechanisms of a cell’s adaptation to low-dose stress.

## 11. Summary and Conclusions

Only few people have heard about the hormesis theory. Nevertheless, hormesis is an important principle in the global biosphere with implications in many fields. This review provides many examples of low-dose stress adaptation in different types of cells. Stressors induce signals in target cells, which are then modulated by cellular response mechanisms to maintain homeostasis and cell survival. One transcription factor (i.e., Nrf2) plays an important role in the modulation of stressors, such as ROS, heat, LDR, Li, and fasting. A protein quality control system involves proteasomes (P), endoplasmic reticulum (ER), and mitochondria (M) (i.e., PERK). Similar control systems exist to protect DNA, RNA, organelles and biomembranes with their lipids. The immune system is another level of protection in multicellular organisms. It protects against infection by microbes and contributes to general homeostasis. Key-lock interactions between ligands and receptors obey rules that are different from linear dose-response relationships.

Without an environmental stress response, archaebacteria from billions of years ago could possibly not have survived the extreme conditions and their manifold changes during evolution. Environmental stress responses fulfil typical criteria, such as stereotypical transcriptional reprogramming and the selection of induced and repressed genes for distinct functions. They also involve epigenetic mechanisms and imprinting, leading to epigenetic memory. This contributes to evolutionary flexibility.

An understanding of hormetic responses and their implications should lead to a shift of paradigm in many fields, for example, radiation biology, toxicology, pharmacology, medicine, marine biology, and agriculture.

Living cells can react to signals from the environment and, thereby, change the rules of linear relationships. The best example is possibly exposure to irradiation. Low doses cause stimulation, for instance, of the immune system, while high doses cause inhibition. The hormesis theory of stress adaptation tries to explain the non-linear dose-response relationship in the global biosphere.

Less can be more. This conclusion holds true for the hormesis effect. With regard to cancer therapy, this statement can also be applied to the development of standard treatments, such as surgery, radio-, and chemotherapy. The dogma of radical, ultra-radical, and supra-radical surgery became replaced by local surgery combined with adjuvant therapy. Aggressive high-dose chemotherapy became transformed, in some instances, into low-dose metronomic application. High-dose radiation apparently works differently to low-dose radiation. The former inhibits while the latter stimulates the immune system. The consequences of this hormesis effect have not yet been implemented into clinical practice. The review described hormetic effects not only of low-dose radiation but also of targeted therapies, oncolytic viruses, and cancer vaccines.

These examples should suffice to change dogmas and paradigms of oncologists. When cancer therapy creates serious adverse events (e.g., of WHO grades 3 to 4), the biological system signals that something goes wrong. When in chronic diseases with multi-organ morbidity, each organ is treated by respective pharmacotherapy, again something goes wrong and creates problems with unpredictable results from multidrug interactions.

The theory of hormesis is still a theory [153,154]. Considerable further research is required to prove, disprove, or modify it. In any case, the theory is important, not only in medicine, but also with regard to the topics agriculture, energy, and environment of our planet [7].

## Figures and Tables

**Figure 1 biomedicines-09-00293-f001:**
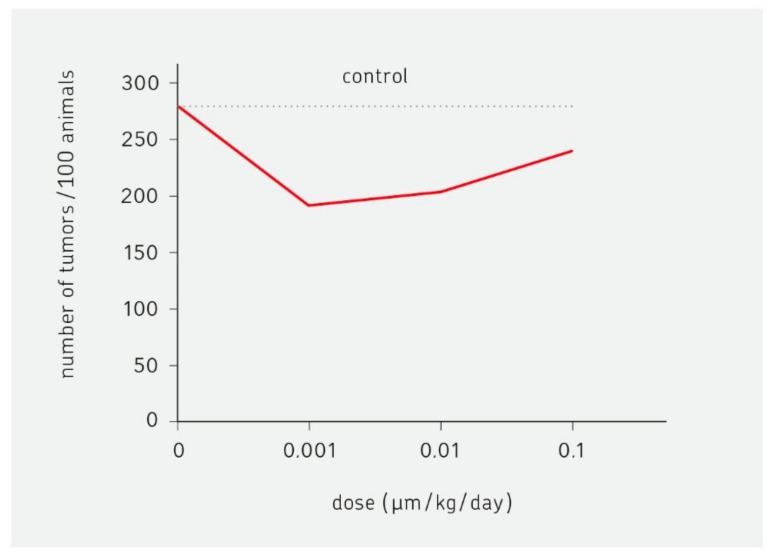
Example of a hormesis effect. The effect of dioxin on the development of breast cancer in rats. In a low dose (0.001 mm/kg/day) the incidence of tumors is strongly reduced. According to Kaiser, J. [46].

**Table 1 biomedicines-09-00293-t001:** Evolutionary aspects of hormesis.

Stress Feature	Inducer/Modulator/Target I/M/T	Mol Mechanism	Effect
Oxidative stress	Glutathion system (M)	TF Nrf2	Homeostasis
Chemodefence	Metals, genotoxics (I)	inducibility	Protection
pH	Sulfonamides (I) QS luxR (M)	Adenyl cyclase	Energy
UV light	Monoterpens (M)	4NQO	UV protection
Radiation	Luminous marine bacteria (T)	3 response levels	Adaptive response
RCS and ROS	PQS of yeast (M)	Mithormesis, glycohormesis	Protection
Fasting	Unicellular to multicellular transformation by Dictyostelium (M)	Polyketide differentiation Inducing factor 1	Reproduction cycle
Fasting	Marine snails (T) Caenorhabditis (worm) (T)	Autophagy SNK-1/Nrf	Reduction of lipofuscin Epigenetic memory

Hormesis is an evolutionary ancient biphasic dose-response of cells and a highly generalizable phenomenon. I = Inducer; M = Modulator; T = Target; RCS = Reactive carbon species; ROS = Reactive oxygen species; QS luxR = Quorum sensing signal receptor; PQS = Protein quality control system; TF = Transcription factor; Nrf2 = Nuclear factor erythroid 2-related factor; 4NQO = Carcinogen 4-nitroquinoline-1-oxide; SNK-1 = Homology to mammalian Nrf2.

**Table 2 biomedicines-09-00293-t002:** Nrf2 and its role in anti-oxidative and anti-inflammatory cellular responses.

Stressor	Response (Part A)	Response (Part B)	Effect
ROS, ER	1. Nrf2 phosphorylation and release from Keap complex 2. Nrf2-P translocation to nucleus	3. Heterodimerization with cMaf, 4. Binding to ARE 5. Transcription of HO-1, NQO1, GCLM	Anti-oxidation
TLR	1. NFκB phosphorylation and release from IKK complex 2. Translocation of NFκB to nucleus 3. Induction of proinflammatory cytokines	4. Induction of HO-1 expression via Nrf2 5. Inhibition of NFκB activation via Nrf2 6. Blocking degradation of IkB-a 7. Degradation of NFκB via Nrf2 8. Inhibition of nuclear translocation via Nrf2	Anti-inflammation

Nrf2 plays a pivotal role controlling the expression of antioxidant genes that ultimately exert anti-inflammatory functions. Nrf2 = Nuclear factor erythroid 2-related factor; Keap = Keap1-Cul3-RBX1 complex; ROS = Reactive oxygen species; ER = Endoplasmic Reticulum; TLR = Toll-like receptor; cMaf = small Maf proteins; HO-1 = Heme oxygenase-1; NQO1 = NADPH quinone oxidoreductase I; GCLM = Glutamate-cystein ligase modifier subunit; NFκB = Nuclear factor kappa B (p50/p65). IKK = Complex between IκB and NFκB; IκB = ankirin repeats-containing NFκB regulatory proteins.

**Table 3 biomedicines-09-00293-t003:** Hormetic effects in the immune system.

Stressor	Sensor/Modulator/Target S/M/T	Mol Mechanism	Effect
LDR	NK cells (S)	p38/MAPK	cytotoxicity
LDR	Macrophages, M1 (S)	iNOS, oxidative burst	Orchestration of T cell immunotherapy
LDR	CD4 and CD8 T cells (S) T regulatory cells (S)	p38/MAPK, NFκB, JNK IL-10 down	Cytokine secretion, CTL activity downregulation
LDR	B cells (S)	NFκB, CD23	OXPHOS shift to aerobic glycosylation
Fungus spore toxin	Drosophila (T)		Increased longevity and fecundity; decreased immune function
Biological threats, infection by microbes	Macrophages (S)	M1/M2 shift	Tissue protection
Transient dietary restriction (DR)	Memory T cells (M), conservation in bone marrow	CXCR4/CXCL12 adipogenesis	Enhanced protective function

The immune system is continuously influenced by hormetic effects of environmental compounds, physical influences and drug and food interactions. S = Sensor, M = Modulator, T = Target; LDR = Low dose radiation; MAPK = Mitogen-activated protein kinase; iNOS = Inducible nitric oxide synthase; NFκB = Nuclear factor kappa B; JNK = c-Jun N-terminal kinase; CXCR4 = Chemokine receptor; CXCL12 = chemokine; OXPHOS = Oxidative phosphorylation.

**Table 4 biomedicines-09-00293-t004:** Clinical implications.

Stressor	Syndrome/Modulator/ Target S/M/T	Mol Mechanism	Effect
Toxic compound Li	Psychiatry (S)	GSK-3, Nrf-2	Stress resistance Longevity
Ag-Nanoparticles (Ag-NPs)	Astroglioma cells (T)	MuD and p38/ERK	Beneficial
Formaldehyde	Bronchial epithelial cells (T)	CyclinD-cdk4, E2F1	Warburg effect
LDR	H_2_O_2_ signaling (M)	Nrf2/Keap1, NFkB	Redox signaling
ROS	Neurodegenerative disorders (S), Curcumin (M) Inflammasomes (T)	Mitochondria, autophagy, apoptosis	Protection
LDR	Autoimmune diseases (S)	Upregulation of Treg Inhibition of cytokines	Regulation of negative effects
H_2_O_2_	NLRP3 inflammasome (T)	PAC1-R	Neuroprotection, neurotrophic and neurogenesis effects
ROS	Cardiovascular diseases (S), MitoPQ (M)	Ca^2+^ homeostasis, mitochondrial homeostasis	Cardioprotection
ROS	Vascular cells (T), EPICAT (M)	Vasodilation	Mitochondrial redox regulation

Exampels of hormetic inducers in a variety of clinical syndroms. Syndrome = Field of clinical implication; M = Modulator; T = Target; LDR = Low dose radiation; NLRP3 = Nucleotide-binding oligomerization domain-like receptor family, pyrin domain-containing inflammasome; ROS = Reactive oxygen species; GSK-3 = Glycogen synthase kinase-3; Nrf-2 = Nuclear factor erythroid 2-related factor; MuD = Mushroom body defect, a microtubule-associated protein that contributes to mitotic spindle function; ERK = Extracellular-regulated protein kinase; CyclinD-cdk4 = CyclinD-cyclin-dependent kinase 4; E2F1 = E2F transcription factor 1; Keap1 = Kelch-like ECH-associated protein 1; PAC1-R = Pituitary adenylate cyclase-activating polypeptide receptor 1; EPICAT = (-)-Epicatechin.

**Table 5 biomedicines-09-00293-t005:** Implications for cancer.

Feature	Inducer/Modulator/Target I/M/T	Mol Mechanism	Effect
Small molecule inhibitor (SMI)	mTOR (T): Aerobic glycolysis, Truncated TCA cycle, MG production (M)	Metabolism of glucose, amino acids, fatty acids, lipids, nucleotides	Targeted inhibition by SMIs of carcinoma growth, MG as hormetin
Oncolytic virus	NDV (I): low-dose optimum for oncolysis, CTL induction and DTH reactivity	HSP27 phosphorylation, proteasomal protein degradation	Oncolysis, Immunogenic cell death (ICD), immune stimulation
SR59230A	ß3-adrenoreceptor (M)	Increase of ROS and cancer cell death	Hormetic low-dose anti-cancer effect
Tumor infiltrating macrophage	Hodgkin lymphoma (T)	CD68+, CD163	Intermediate numbers associated with better prognosis
LDR	Cancer and ulceratice colitis (T)	Radiation hormesis	Three case reports of positve effects
Radon	Cancer (T), primary or adjuvant treatment	Radiation hormesis	Four case reports of positive effects

Examples of hormetic inducers in cancer therapy. mTOR = Mammalian target of rapamycin; I = Inducer; M = Modulator; T = Target; NDV = Newcastle disease virus; LDR = Low-dose radiation; OXPHOS = Oxidative phosphorylation; TCA = Tricarbonic acid cycle; MG = methylglyoxal; CTL = Cytotoxic T lymphocyte; DTH = Delayed-type hypersensitivity; HSP27 = Heat-shock protein 27.

**Table 6 biomedicines-09-00293-t006:** Herbicid hermetic effects in plants.

Herbicid	Modulator/Target M/T	Mol Mechanism	Effect
Metal: Cd or Pb	ROS (M)	Increase in auxin andflavonol	Hormetic stimulation of shoot growth
Metal: Ag-NP	Maize (T)	Positive effect on plants roots	Negative effect on rhizome
Glyphosate, 2,4-D, Paraquat	ROS (M)	H_2_O_2_ as signaling molecule	Increased water transport causing cell expansion
Silicon (Si)	Si accumulators: rize, wheat, barley, sugarcane, soybean, sugarbeet (T)	Si binding to hydroxyl groups of proteins involved in signaling	Hormetic effect on growth, chlorophyll, amino acids and sugars

Examples of herbicides with hormetic effects. M = Modulator; T = Target; Cd = cadmium; Pb = lead; Ag-NP = silver-nanoparticles; ROS = Reactive oxygen species; H_2_O_2_ = Hydrogenperoxide.

**Table 7 biomedicines-09-00293-t007:** Archaic environmental stress response.

Environmental Stressor	Species/Genes/Transcription	Response Criterium
Heat shock 25 °C to 37 °C	Halobacterium salinarum iESR: 724 genes rESR: 276 genes	1. Global, stereotypical transcripttional reprogramming
Heat shock 25 °C to 37 °C	H. salinarum Repression of genes involved in ribosome biosynthesis and translation	2. Induced and repressed genes enriched for distinct functions
Paraquat Redox cycling agent	H. salinarum 4 mM caused higher response than 0.25 mM	3. Duration and magnitude of the transcriptional response dendent on intensity of stress
Reciprocal environmental shift 37 °C to 25 °C	H. salinarum Rapid recovery from stress without ESR-like transcriptional characteristics	4. Induction of the transcriptional response specific to stress exposure

iESR = Induced environmental stress response; rESR = Repressed environmental stress response.

**Table 8 biomedicines-09-00293-t008:** Stress adaptation in the global biosphere.

Stress Type	Example	Mol Mechanism	Effect
Physical	Temperature (heat, frost), radiation, exercise	Nrf2, glutathion DNA methylation, microRNA	Cell protection Apoptosis, autophagy, cell cycle regulation, DNA repair and turnover
Chemical	ROS, Li, Si, Ag, Cd, Pb Monoterpenes Methylglyoxal	Proteasome, endoplasmic reticulum, PQS, mitochondria	Cell survival Epigenetic memory Evolutionary flexibility
Biochemical	Dietary restriction, Pharmacological drugs	Glucose-ketone switch PAC1-R	Metabolic switching Neuroprotection
Biologic	Hormone: melatonin Oncolytic virus: NDV	Calcium HSP27, Type I IFN	Adaptation to circardian rhythm Oncolysis, Immunogenic cell death

NDV = Newcastle disease virus; Nrf2 = Nuclear factor erythroid 2-related factor; PQS = Protein quality control system; PAC1-R = Pituitary adenylate cyclase-activating polypeptide receptor 1; HSP27 = Heat-shock protein 27; IFN = Type I interferon.

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
