# Peer review of "Less Can Be More: The Hormesis Theory of Stress Adaptation in the Global Biosphere and Its Implications"

_biomedicines, 2021, doi:10.3390/biomedicines9030293_

Round 1
Reviewer 1 Report
The present review describes hormesis as an (actually still unvalued) principle underlying the responses of cells and organisms to environmental stress. The author offers ample valuable information on hormetic responses of selected biological systems. As a core focus the review deciphers hormetic reactions of special relevance in medicine. The reviewer strongly supports publication of the manuscript.
Some comments and suggestions might be considered by the author for minor revision of the script.
Comments
Page 1 line 7
“A dose-response relationship to stressors, according to the hormesis theory, is characterized by low-dose stimulation and high-dose inhibition. “
The dimension of stress responses e.g. vitality or fitness should be added.
Page 1 line 25
“The effects of hormesis during evolution are cell protection and survival, evolutionary flexibility, and epigenetic memory.”
These “effects of hormesis” seem to be somewhat arbitrary and unrelated. Might be proofed and changed.
General point
To further improve readability of the review the author might consider addition of some cartoons illustrating hormesis definition and some exemplary hormetic responses.
Author Response
Thank you for the positive response.
The points at page 1 (line 7 and 25) have been delt with accordingly.
As requested, a figure has now been added with corresponding text lines 266-272.
Thank you for the positive response.
Reviewer 2 Report
Authors described a new review and implications of horemesis. The descriptions are very important and contents are very useful. However, authors should deal with several minor points.
1) In all tables, two subjects are both above and below tha Tables. Both should be integrated and descrived above Tables.
2) In all tables, references' numbers should be added.
3) There are two Table 7s. The latter Table 7 must be changed to Table 8.
4) The 1st and 2nd columns of Tables 1, 3, 4, 5, 6 and 8 are difficult to be understood for readers. Therefore, instead of each 1st and 2nd column presented as words, authors should integrate the 1st and 2nd column, and should discribe a sentence.
That is, "(any sentence)", "Mol Mechanism", "Effect" should be expressed.
5) In Table 2, it is difficult for the reader to understand that "Response (Part A)" is divided into "Response (Part B)". Therefore, "Response (Part A)" and "Response (Part B)" should be integrated to "Response".
6) In page 13, authors wrote "The dose-response curve was bell-shaped like in hormesis [87]." However, I can not understand what part in [87] is bell-shaped? Is this sentence correct? If it is not correct, authors must change it.
7) In page 23, "11. Summary" and "12. Conclusions" should be integrated.
8) In the last paragraph of page 24, authors wrote "The theory of hormesis is still a theory". This sentence should have references, and must be changed to "The theory of hormesis is still a theory [6, 151, 152]". The references 151 and 152 are as follows.
151. Kino, K. The prospective mathematical idea satisfying both radiation hormesis under low radiation doses and linear non-threshold theory under high radiation doses. Genes Environ. 2020, 42, 4. Doi: 10.1186/s41021-020-0145-4.
152. Devic, C.; Ferlazzo, M.L.; Berthel, E.; Foray, N. Influence of individual radiosensitivity on the hormesis phenomenon: Toward a mechanistic explanation based on the nucleoshuttling of ATM Protein. Dose Response. 2020, 18, 1559325820913784. Doi: 10.1177/1559325820913784.
Author Response
Thank you fort he positive response.
1) The text above the Tables has been set by the Editor, not the author. The author
has set the last sentence back tot he corresponding paragraph.
2) For the tables the author used the Word table format. He tried to condense as much as possible
into this given structure. More text is not possible without destroying the format. References are
available in the main text.
3) This has been corrected.
4) See response to point 2. The author has had hard times to optimize the tables and does not dare
to change anything. In former times this job has been with the Editors.
5) See response to point 2 and 4.
6) The author excuses himself to the reviewer. The quotation was wrong. It is now ref. 90 by Cassel
and Garrett.
A viral oncolysis in the peritoneal cavity was obtained only when an intermediate dose of virus
was used. Lower and higher doses were ineffective. That makes the bell-shaped curve.
7) Summary and Conclusions have been combined.
8) The two new references have been added.